# An Efficient GPU Implementation of a Coupled Overland-Sewer Hydraulic Model with Pollutant Transport

**Javier Fernández-Pato** [1,2,*,†,‡] **and Pilar García-Navarro** [1,‡]

1   I3A, Fluid Mechanics, University of Zaragoza, 50018 Zaragoza, Spain; pigar@unizar.es
2   Hydronia-Europe S.L., 28046 Madrid, Spain
*   Correspondence: jfpato@unizar.es; Tel.: +34-876555057
†   Current address: Ed. Torres Quevedo, María de Luna 3, CP, 50018 Zaragoza, Spain.
‡   These authors contributed equally to this work.

**Abstract:** Numerical simulation of flows that consider interaction between overland and drainage networks has become a practical tool to prevent and mitigate flood situations in urban environments, especially when dealing with intense storm events, where the limited capacity of the sewer systems can be a trigger for flooding. Additionally, in order to prevent any kind of pollutant dispersion through the drainage network, it is very interesting to have a certain monitorization or control over the quality of the water that flows in both domains. In this sense, the addition of a pollutant transport component to both surface and sewer hydraulic models would benefit the global analysis of the combined water flow. On the other hand, when considering a realistic large domain with complex topography or streets structure, a fine spatial discretization is mandatory. Hence the number of grid cells is usually very large and, therefore, it is necessary to use parallelization techniques for the calculation, the use of Graphic Processing Units (GPU) being one of the most efficient due to the leveraging of thousands of processors within a single device. In this work, an efficient GPU-based 2D shallow water flow solver (RiverFlow2D-GPU) is fully coupled with EPA's Storm Water Management Model (SWMM). Both models are able to develop a transient water quality analysis taking into account several pollutants. The coupled model, referred to as RiverFlow2D-GPU UD (Urban Drainge) is applied to three real-world cases, covering the most common hydraulic situations in urban hydrology/hydraulics. A UK Environmental Agency test case is used as model validation, showing a good agreement between RiverFlow2D-GPU UD and the rest of the numerical models considered. The efficiency of the model is proven in two more complex domains, leading to a >100x faster simulations compared with the traditional CPU computation.

**Keywords:** finite volumes; shallow-water equations; RiverFlow2D-GPU; SWMM; water quality

---



## 1. Introduction

Urban areas usually have a high concentration of population, business and infrastructures, representing extremely vulnerable regions to natural hazards such as floodings, which can lead to important social and economic losses [1–5]. Flood risk management has historically focused on coastal and fluvial flooding, with less emphasis on urban surface water flooding [6]. Nevertheless, the advance of urbanization over green areas leads to a soil impermeabilization and the limited discharge capacity of the drainage network of the cities has increased runoff volumes and flooding in significant areas. Another important fact to take into consideration is that road and railroad infrastructure constitute barriers which interfere with surface runoff. There are a significant number of studies reporting how the socio-economic losses linked to urban floodings are expected to increase in the forthcoming decades, especially in regions with an emergent economic development or extreme weather conditions [1,7,8].

All these potential phenomena reveal the need for developing contingency plans for the vulnerable regions in order to be able to make decisions quickly and in an orderly

and smart way. In this sense, the numerical simulation of flows that consider interaction between overland and drainage networks has become an essential tool to predict and prevent floodings, specially when dealing with intense storm events, which represent challenging conditions for drainage capacity of the sewer systems. The interaction between the sewer network and the overland flow deserves to be carefully designed [9]. It should be bidirectional and the exchange flow should be computed in terms of both the values of the surface water depth and the pressure head of the pipe at the sewer-surface linking points.

During the last few decades, many different models for surface flow have been developed. Due to the irregular nature of streets and buildings network in a urban environment, two-dimensional (2D) flow models are usually the most suitable for this type of simulation. Two main approaches are considered for this purpose: Shallow Water-based models (SW) [10–15] and Diffusion-Wave of Zero-Inertia (ZI) models, which represent a simplification of the SW models [16–21]. Interesting comparisons between 1D and 2D choices for modeling the surface flow are presented in Leandro et al. [22] and Leandro et al. [23], clarifying the limitations of the 1D surface flow models for urban environments and emphasizing that the correct modelization of the surface pathways and the linking elements between both domains (manholes) are key factors to set up an accurate surface–sewer flow coupled model. Currently, the increase in the computational power of personal computers has caused 2D SW-based models to become increasingly popular due to their accuracy, especially when considering highly transitory cases. On the other hand, water flow in sewer systems is usually modelled by means of 1D models to route the water flow [19,24–27]. A very recent review of the current urban surface water flood models is presented in Guo et al. [6].

A remarkable effort to model the pollutant transport has been made during the past few decades in order to efficiently follow up potential contaminant releases in natural and urban areas [28]. As in the water flow simulation, recent developments in computational modelling of the contaminant transport allow for accurate 2D simulation of transient loading over complicated geometries [28–32]. The pollutant transport within the sewer system is carried out by means of a scalar equation additional to the water flow routing [25,26,33,34]. The level of complexity of the pollutant transport model is also a factor to take into account, ranging from a simple passive transport of a single or several substances [28,31] to a full water quality model as the WASP (Water Quality Analysis Simulation Program) model [35], which considers 10 extra depth-averaged transport equations (one for each water quality state variables considered in the model), as well as 20 conversion processes among them, leading to a complex reaction matrix.

When considering realistic large domains with complex topography or roads/streets structure, a fine spatial discretization is mandatory in order to obtain an accurate representation of these elements, hence usually requiring a very large number of computational cells. Therefore, an extra requirement to perform an efficient simulation is the use of parallelization techniques, the use of Graphic Processing Units (GPU) being one of the most efficient due to the leveraging of thousands of processors within a single device. The use of GPU devices for hydraulic simulations allows us to obtain high computation accelerations with respect to a single-core CPU [36–38].

In this study, an efficient GPU-based 2D Shallow Water (SW) flow solver (RiverFlow2D-GPU) is fully coupled with EPA's Storm Water Management Model (SWMM) [25,26]. The coupled model, referred as RiverFlow2D-GPU UD (Urban Drainage) is applied to three real-world cases, covering the most common hydraulic situations in urban hydrology/hydraulics. To the authors knowledge, no previously published works consider either GPU computing for a fully coupled SW-based surface flow connected with a drainage model or pollutant transport calculations for this kind of model.

## 2. Mathematical Model

### 2.1. 2D Surface Flow Model

Overland flow is modeled through the 2D shallow-water equations [39], augmented with the depth averaged transport equation forming a system which can be written as follows:

$$\frac{\partial \mathbf{U}}{\partial t} + \frac{\partial \mathbf{F}(\mathbf{U})}{\partial x} + \frac{\partial \mathbf{G}(\mathbf{U})}{\partial y} = \mathbf{S} + \mathbf{H} + \mathbf{M} + \mathbf{P} \tag{1}$$

where

$$\mathbf{U} = \left( h, q_x, q_y, h\phi \right)^T \tag{2}$$

are the vector of conserved variables, where $h$ represents the water depth and $q_x = hu$ and $q_y = hv$ are the unit discharges, with $u$ and $v$ the depth averaged components of the velocity vector $\mathbf{u}$ along the $x$ and $y$ coordinates, respectively. $\phi$ is the depth averaged solute concentration. The fluxes of these conserved variables can be written as

$$\mathbf{F} = \left( q_x, \frac{q_x^2}{h} + \frac{1}{2}gh^2, \frac{q_x q_y}{h}, h\phi u \right)^T \tag{3}$$

$$\mathbf{G} = \left( q_y, \frac{q_x q_y}{h}, \frac{q_y^2}{h} + \frac{1}{2}gh^2, h\phi v \right)^T \tag{4}$$

where $g$ is the acceleration due to gravity. The source terms of (1) are split into four addends. The term $\mathbf{S}$ represents the friction losses defined as

$$\mathbf{S} = \left( 0, -ghS_{fx}, -ghS_{fy}, 0 \right)^T \tag{5}$$

where $S_{fx}, S_{fy}$ are the friction slopes in the $x$ and $y$ direction, respectively, here expressed in terms of the Manning's roughness coefficient $n$:

$$S_{fx} = \frac{n^2 u\sqrt{u^2 + v^2}}{h^{4/3}}, \qquad S_{fy} = \frac{n^2 v\sqrt{u^2 + v^2}}{h^{4/3}} \tag{6}$$

The term $\mathbf{H}$ accounts for the pressure force variation along the bottom in $x$ and $y$ directions and can be formulated in terms of the bottom level $z$ bed slopes:

$$\mathbf{H} = \left( 0, -gh\frac{\partial z}{\partial x}, -gh\frac{\partial z}{\partial y}, 0 \right)^T \tag{7}$$

The term $\mathbf{M}$ represents the mass sources/sinks due to rainfall/infiltration:

$$\mathbf{M} = (R - f + f_M, 0, 0, 0)^T \tag{8}$$

where $R$ is the local rainfall intensity, $f$ the infiltration rate, computed by means of the Curve Number method [40,41] and $f_M$ refers to a discharge per unit area of the manhole mouth. There is a finite number of locations where the latter term participates, hence it has been formulated as follows:

$$f_M = \sum_j I_j(x,y)\frac{Q_{e,j}}{A_{M,j}} \tag{9}$$

where $I_j(x,y) = 1$ if point $(x,y)$ corresponds to a manhole location and it is nil otherwise. $Q_e$ is the exchange discharge between both domains and $A_M$ is the area of the manhole.

Regarding the solute or pollutant source term $\mathbf{P}$, it can be expressed as a function of the diffusion-dispersion tensor $\mathbf{K}$:

$$\mathbf{P} = (0, 0, 0, \nabla(\mathbf{K}h\nabla\phi))^T \tag{10}$$

It is feasible to neglect molecular difussion, expressing the tensor **K** as a diagonal matrix, only taking into account turbulent dispersion velocity terms:

$$\begin{pmatrix} K_L & 0 \\ 0 & K_T \end{pmatrix} \tag{11}$$

where $K_L$ nad $K_T$ represent the longitudinal and transversal diffusion coefficients, respectively. In general, $\phi$ can be a vector of constituents in which case additional concentration variables should be added to the the different terms of Equation (1) so that each pollutant is transported by a separate equation.

Assuming dominant advection, (1) can be classified as hyperbolic system. In this case, the Jacobian matrix of the flux can be diagonalized and three real eigenvalues can be found. They are used to build the numerical contributions that update the flow variables and to control the numerical stability of the method. The system can be solved by an explicit, first-order, upwind finite volume scheme described in [42]. The wet/dry fronts are well tracked providing stable solutions with a numerical mass error similar to machine accuracy. Adequate boundary conditions must be imposed and the procedure also requires initial conditions.

### 2.2. 1D Pipe Flow Model

In this work, pipe flow routing is modeled by means of the routing portion of SWMM 5.1 software [25,26], which transports the surface inflows through a user-designed system of pipes conforming a network of links connected together at nodes. The routing of the water flow is governed by the 1D St. Venant equations of conservation of mass and momentum:

$$\frac{\partial A}{\partial t} - \frac{\partial Q}{\partial x} = q_e \tag{12}$$

$$\frac{1}{A}\frac{\partial Q}{\partial t} + \frac{1}{A}\frac{\partial}{\partial x}\left(\frac{Q^2}{A}\right) + g\frac{\partial h_p}{\partial x} - g\left(S_0 - S_f\right) = 0 \tag{13}$$

where $A$ is the flow cross-sectional area, $Q$ is the flow rate, $h_p$ is the conduit water depth and $q_e$ is the exchange discharge $f_M$ per unit lenght.

The transport of dissolved substances along the lenght of a conduit is modeled in SWMM by the following mass conservation equation:

$$\frac{\partial \phi}{\partial t} = -\frac{\partial u\phi}{\partial x} + \frac{\partial}{\partial x}\left(D\frac{\partial \phi}{\partial x}\right) + r(\phi) \tag{14}$$

being $u = Q/A$ the longitudinal velocity, $D$ the longitudinal dispersion coefficient and $r(\phi)$ the reaction rate term. As in the 2D model, $\phi$ can be a vector of constituents in which case a separate Equation (14) would apply for each pollutant. In this case, the reaction term could be a function of several constituents.

The solution method for (12) and (13) involves a finite difference discretization with an implicit backwards Euler method in order to provide additional numerical stability (Rossman (2017)).

### 2.3. Water Exchange between Models

Several situations regarding the flow exchange between the surface flow and the sewer system can take place. Figure 1 displays all the possible scenarios: (1) Inflow into non-pressurized sewer, (2) Inflow into pressurized sewer, (3). Outflow over floodplain (wet or dry). Every time step, an internal algorithm compares the values of the surface water depth ($h$), pressure head in the pipe ($h_p$) and the distance between the bed of the flume and the invert level of the sewer ($H = z + (z_p + H_{max})$), in order to adequately estimate the exchange discharge in terms of the diameter of the manhole $D_M$, area of the manhole $A_M$, and a coefficient $C$ which accounts for the energy losses at the manhole. The particular form

used to formulate the exchange discharge $Q_e$ follows closely the formulation suggested in [43].

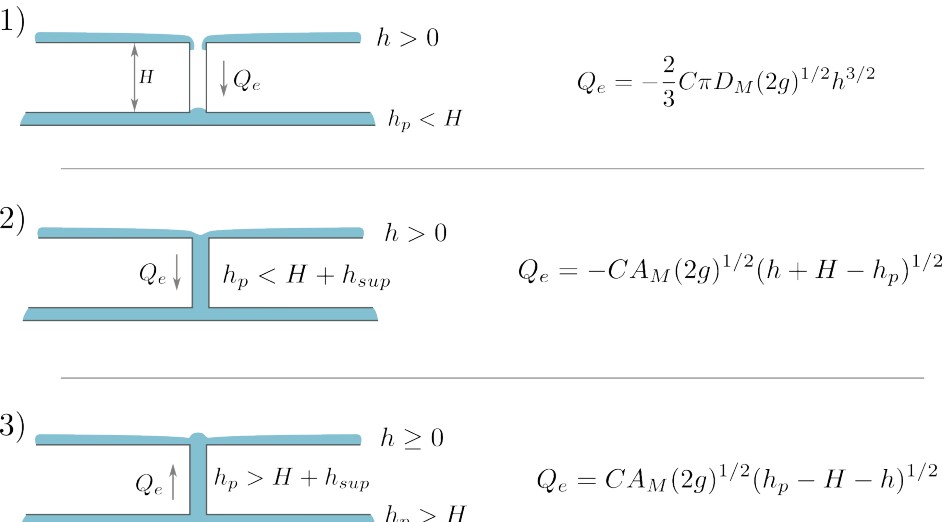

$$Q_e = -\frac{2}{3}C\pi D_M (2g)^{1/2} h^{3/2}$$

$$Q_e = -CA_M (2g)^{1/2}(h + H - h_p)^{1/2}$$

$$Q_e = CA_M (2g)^{1/2}(h_p - H - h)^{1/2}$$

**Figure 1.** Possible hydraulic scenarios in the coupled model.

When coupling two or more numerical models, it is necessary to take into account that the time steps may be different. Therefore, it is necessary to homogenize this value so that the models work in a synchronized way. In this case, since SWMM routing algorithm is based on an implicit method, it has no restrictions on the time step and the surface model (explicit) governs the temporal advance of the simulation, due to stability issues.

## 3. Test Cases

### 3.1. Case 1: UK Environmental Agency Case 8b

This case, presented in Néelz and Pender (2013), tests the model capability to simulate shallow inundation originating from a surcharging underground pipe. The pipe is modelled in 1D and connected to the 2D grid through a manhole. The modelled area is approximately 0.4 km by 0.96 km (see Figure 2 for the bed elevations). A culverted watercourse of circular section of 1.4 m in diameter and 1340 m in lenght is assumed to run through the modelled area. The pipe Manning's roughness is set to $n = 0.017$. An inflow discharge boundary condition is applied at the upstream end of the pipe, illustrated in Figure 3 (left). A free outfall is considered as downstream boundary condition. A steady state corresponding to a baseflow of 1.6 m$^3$/s is set as initial condition. A surcharge is expected to occur at a vertical manhole of 1 m$^2$ located 467 m from the top end of the culvert at the coordinates P1($x$ = 264,896 m, $y$ = 664,747 m). The sewer outlet is located at the point O1($x$ = 271,874.4, $y$ = 661,752.8 m). The profile geometry of the culvert is shown in Figure 3 (right). The overland spatial domain is discretized using an unstructured triangular mesh of 95,784 cells.

Figure 4 shows the numerical results for water level elevation at probes 1 and 7. Comparison between CPU and GPU computations is also depicted, showing a perfect match between them. The results mainly fall within the envelope of the numerical results generated by the rest of the hydraulic models provided in Néelz and Pender (2013), which is an indicator of the good behaviour of the coupled model in the absence of observed data. Regarding the computational efficiency, the GPU simulation was 25× faster than the CPU computation, for this particular case.

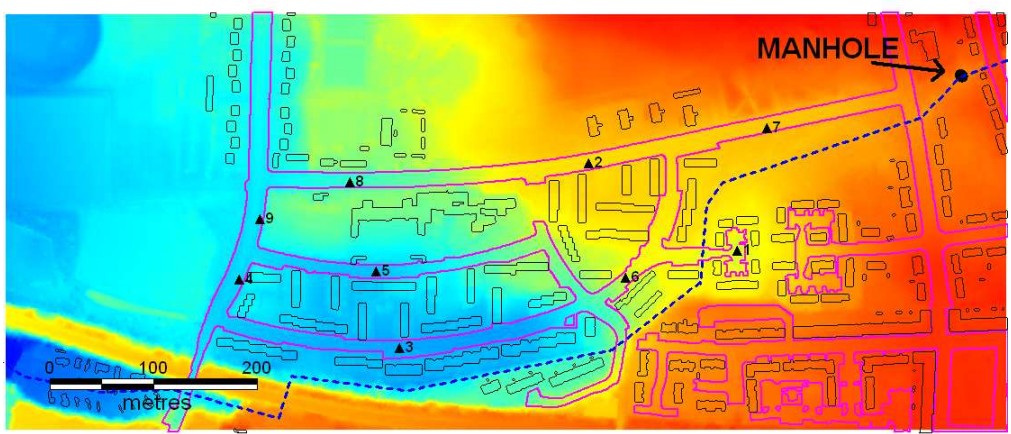

**Figure 2.** Case 1: DEM used, with the location of the manhole. The course of the underground pipe is indicated, although irrelevant to the modelling. Purple lines: outline of roads and pavements. Black lines: building outlines. Triangles: output point locations.

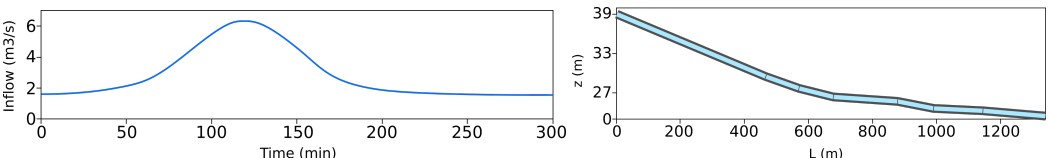

**Figure 3.** Case 1: Inflow hydrograph applied at upstream end of culvert (**left**). Culvert profile geometry (**right**). $z$ refers to the pipe elevation over el outlet point.

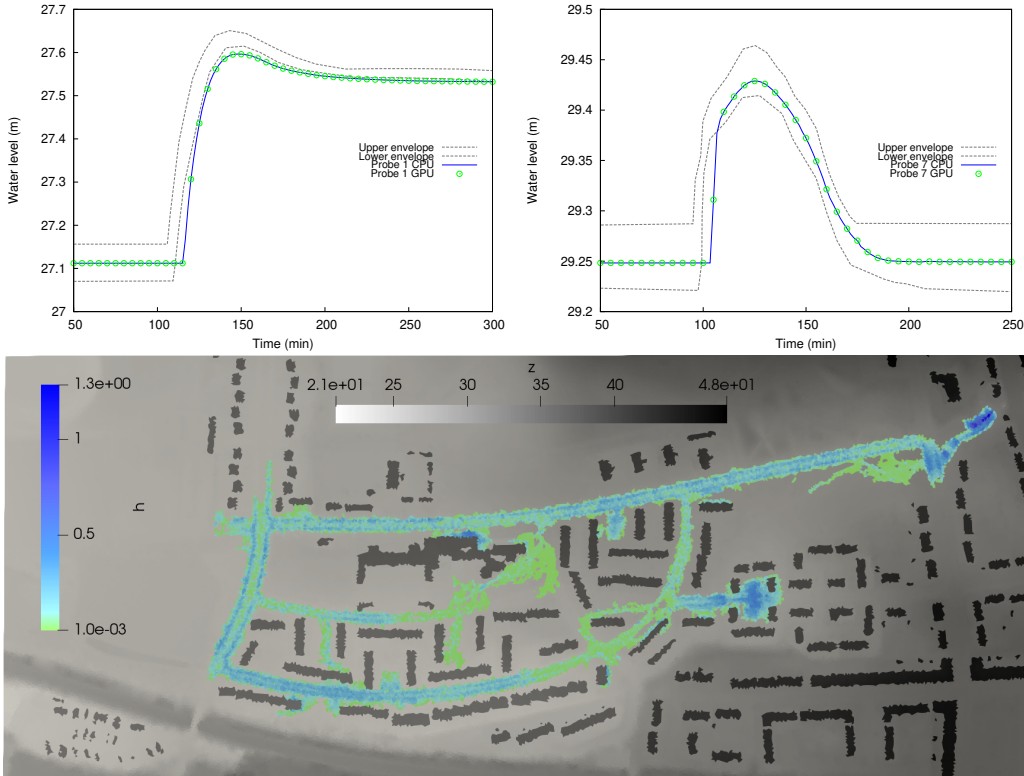

**Figure 4.** Case 1: Water level evolution at probes 1 (**upper**, **left**) and 7 (**upper**, **right**). Simulated flood extent at final time.

In order to test the capacity of the model to transport and exchange pollutants through both sewer and surface domains, the base configuration is modified by adding two pollutant inflows within the sewer network. Solids are one of the most common contaminants

found in urban storm water, hence a total suspended solid (TSS) concentration of 25 mg/L during 5 h is set in the inflow boundary condition. On the other hand, a lead injection of 25 mg/L for 5 h is considered together with a constant discharge inflow of 1.2 m³/s at the coordinates P2 ($x$ = 264,730.5 m, $y$ = 664,602.9 m). Additionally, a second manhole of 1 m² cross-section located 991 m from the inlet of the culvert at the coordinates ($x$ = 264,735.5 m, $y$ = 664,601.9 m) is opened. For the sake of clariry in the nomenclature, from now on, $\phi_1$ and $\phi_2$ will refer to the concentration of TSS and lead, respectively.

Figures 5 and 6 show the water and pollutants flood extent at $t$ = 1.8 h (left column) and $t$ = 5 h (right column). Several facts are noteworthy in these results. At $t$ = 1.8 h (Figure 5), both inflows supply water to the surface (as well as each own pollutant) at P1 and P2 locations but both surface flows are disconnected yet. On the other hand, the sewer pipe has routed the $\phi_1$ sewer inflow to P2 manhole, reaching the surface due to the pressurization of the pipe at this point (Figure 5, center). As it can be seen in Figure 5 (lower), $\phi_2$ affects only the areas downstream the P2 location. At $t$ = 5 h (Figure 6), both sewer inflows have ended and the final flood and pollutants extents are shown, leading to a maximum water depth values of $h$ = 1.5 m and maximum concentrations of $\phi_1$ = 23 mg/L for TSS and $\phi_1$ = 9.5 mg/L for lead.

The water depth and pollutant concentration values can also be monitorized at the sewer system by representing the values provided by SWMM against time. Figure 7 shows the temporal evolution of sewer water depth and pollutant concentrations at the inflow point P1 (upper) and at the sewer outlet O1.

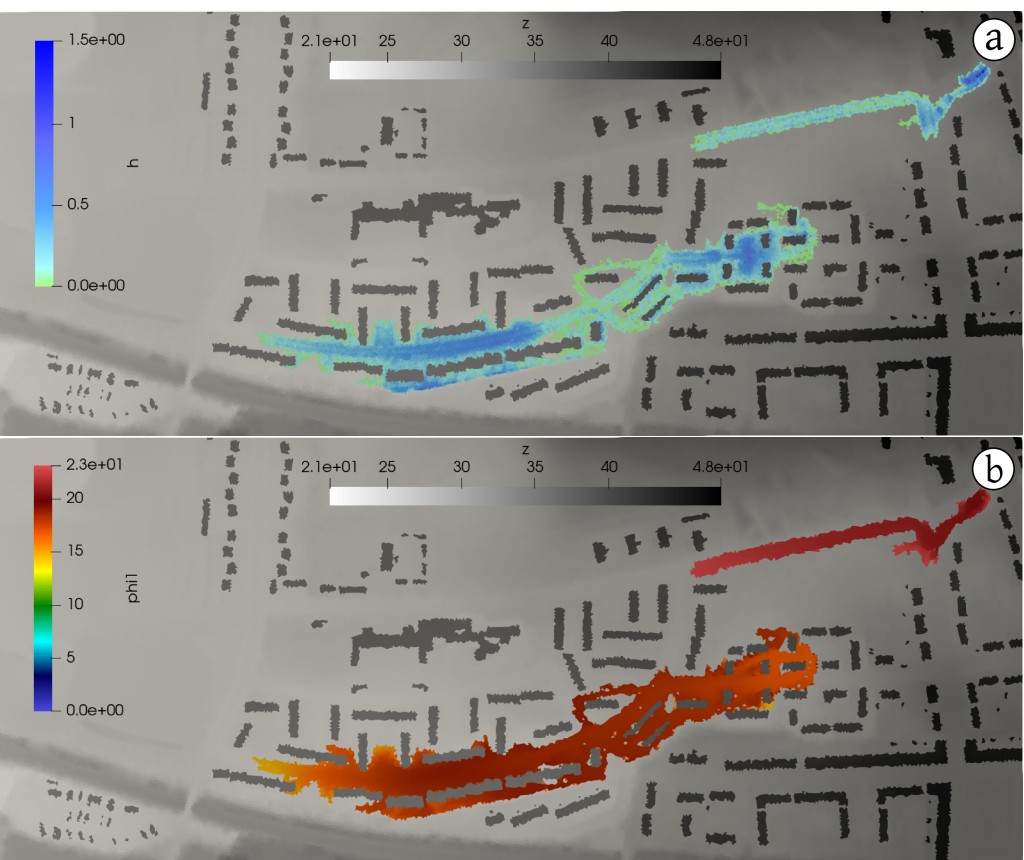

**Figure 5.** *Cont.*

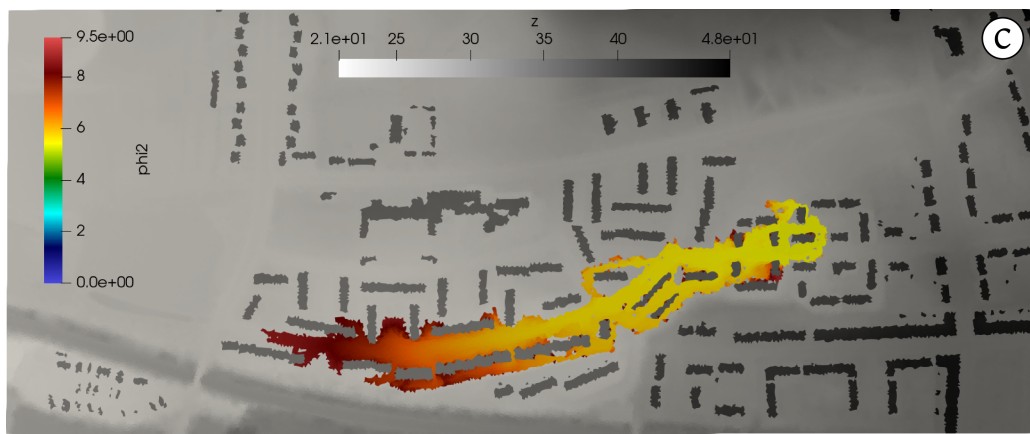

**Figure 5.** Case 1: Water (**a**) and pollutants (**b**,**c**) flood extent at $t = 1.8$ h.

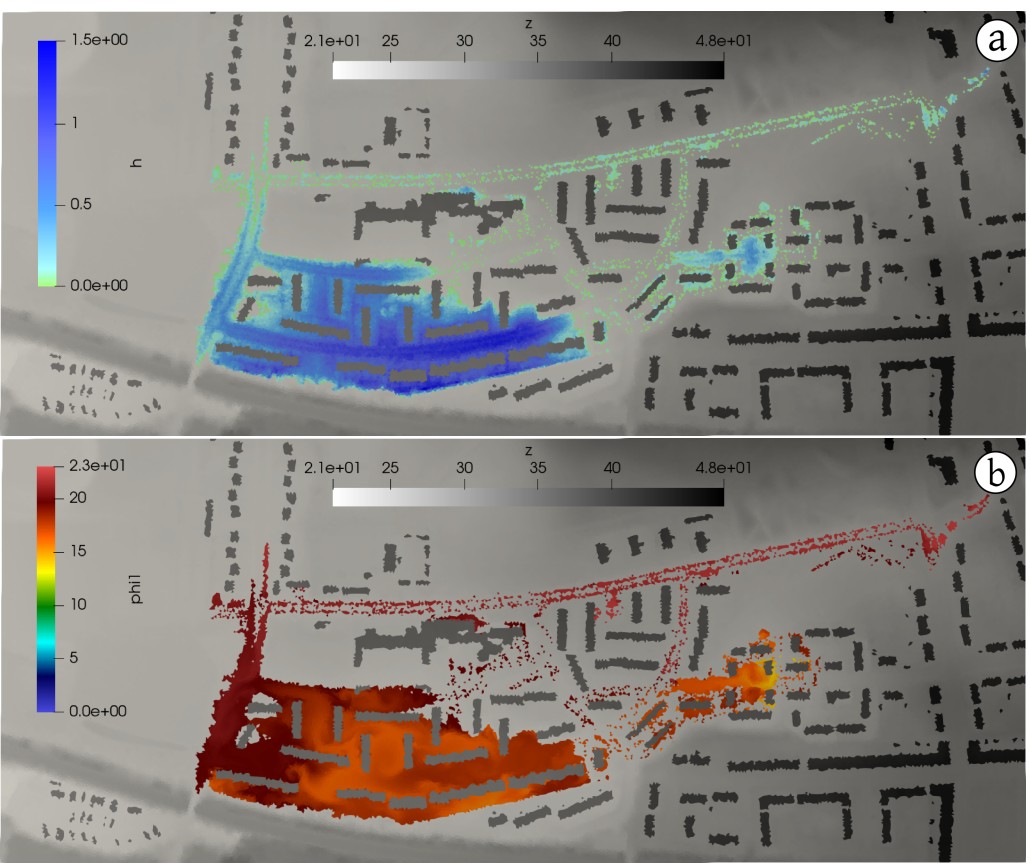

**Figure 6.** *Cont.*

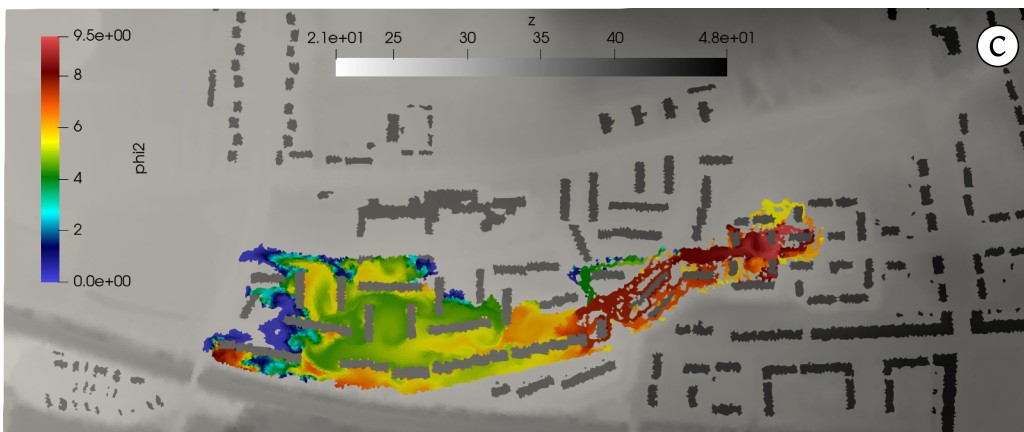

**Figure 6.** Case 1: Water (**a**) and pollutants (**b**,**c**) flood extent at $t = 5$ h.

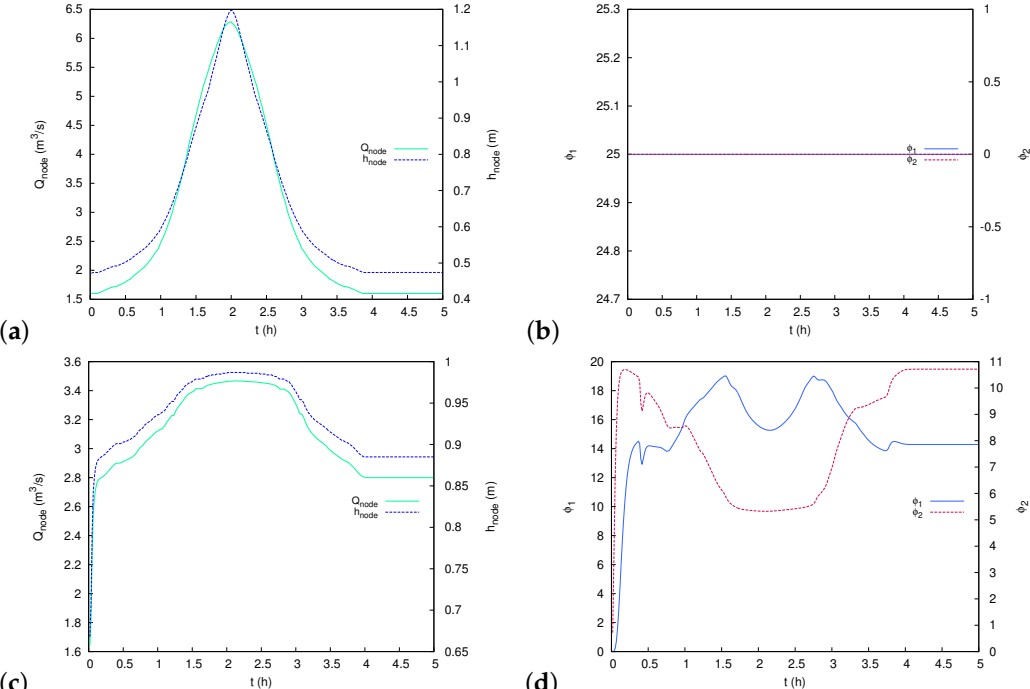

**Figure 7.** Case 1: Temporal evolution of sewer water depth (**a**,**c**) and pollutant concentrations (**b**,**d**) at the inflow point P1 (**upper**) and at the sewer outlet O1 (**lower**), respectively.

### 3.2. Case 2: Storm Drainage Capacity Test

In this case, a uniform rainfall is set all over an initially dry complex urban topography. An underground drainage network has been considered with 36 junctions/surface connections and 40 linking pipes. Figure 8 shows the bed elevations map together with the pipe system (left) and a detail of the unstructured triangular computational mesh of 169,333 cells, carefully refined in order to preserve the complex inner geometry of the domain (right). For the sake of simplicity, a uniform Manning's roughness coefficient of 0.02 is set for the surface domain. No pullutants are considered in this test case.

The drainage network is initially empty and all the pipes are single-barrel and circular with a constant diameter of 1 m and a uniform Manning's roughness of $n = 0.01$. This is a small value for a sewer network, but it is not relevant for the purpose of this test case, which is to evaluate the efficiency of GPU parallelization. All the connecting nodes have a manhole diameter of 0.8 m and a maximum height of 1 m. The difference in elevations between the highest and the lowest node is 5 m. The longest segment is 200 m and the shortest is 15 m.

The purpose of this test case is two-fold. On the one hand, the objective is to evaluate the ability of the model to drain water from an intense storm (25 mm/h during 24 h) and check if a water upwelling occurs at any point of the network. On the other hand, the simulation will be carried out with three different versions of the code: CPU (one core), CPU (six cores) and GPU. In this way, the efficiency of the GPU can be tested in a more suitable computational mesh with a large number of cells.

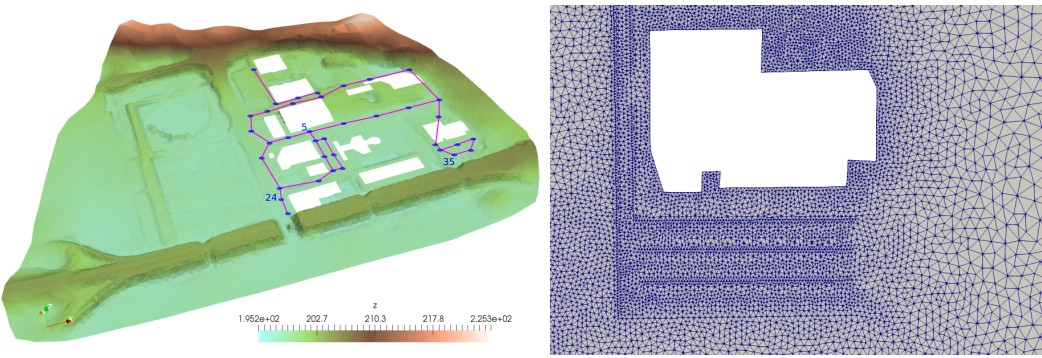

**Figure 8.** Case 2: 3D representation of the domain topography (**left**) and detail of the computational mesh showing the local refinement (**right**).

Figure 9 shows the temporal evolution for the water depth and flooding discharge in the selected nodes (5, 24 and 35) at strategic locations (see Figure 8 (left)). The top level of the pipe is also represented. Nodes 5 and 24 remain unpresurized during all the simulation. On the other hand, node 35 is overflowing after the first hour of simulation. This node can be considered representative of the lowest part of the network, which remains continuously overflowing during the rest of the simulation with maximum discharge peaks of 0.2 m$^3$/s.

In order to test the efficiency of the GPU model under highly transient conditions and a large number of wet cells, the computational time has been measured for CPU (one core), CPU (six cores) and GPU setups (Table 1). The speed-up factor is also shown, defined as the ratio between the simulation time in CPU with a single core and the simulation time of a multi-core or GPU simulation. The CPU simulations have been performed using an Intel Core i7-8700 at 3.70 Ghz processor, whereas NVIDIA Tesla C2075 and NVIDIA GTX Titan Black devices have been used to run the GPU simulations.

### 3.3. Case 3: Pollutant Transport in a Mixed Environment

The purpose of this case is to test the ability of the coupled model to deal with a combination of several hydraulic/hydrologic phenomena over a huge and complex domain. The study area is located in the city of Santa Fé (Argentina). Figure 10 (upper) shows the urban map and the delimitation of the study domain (red polygon). Several areas of interest with very different characteristics are considered including a large and complex network of streets and roads, a golf course and an artificial lake connected to an important river (Río Salado) upstream and to a supply channel downstream.

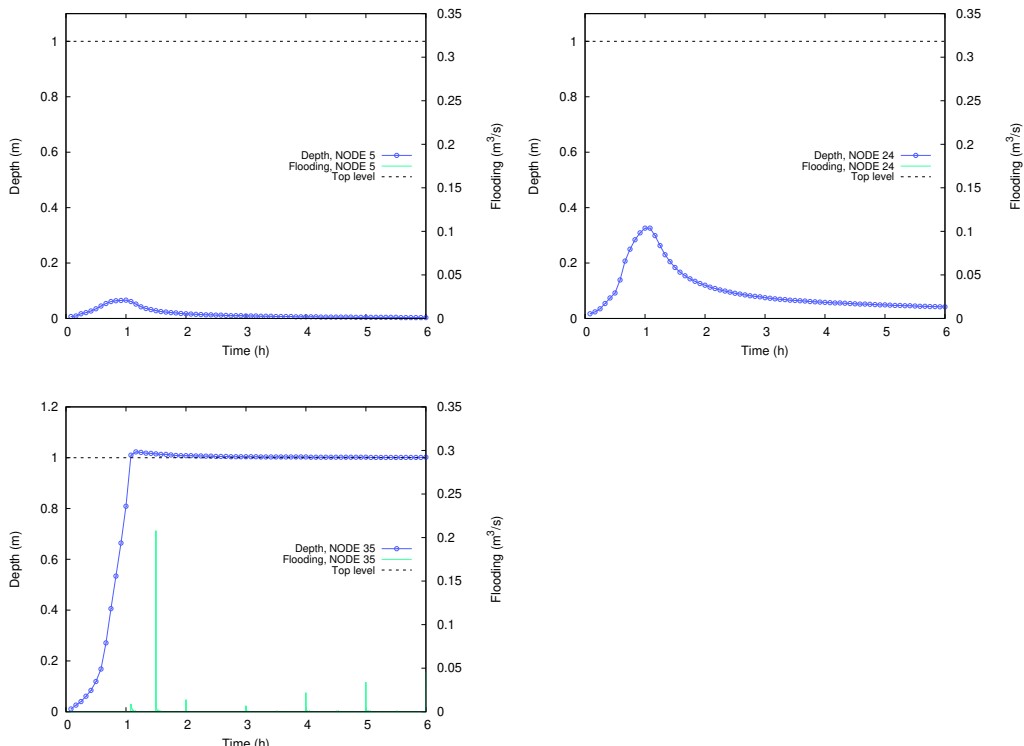

**Figure 9.** Case 2: Temporal evolution of the pipe water depth and flooding discharge at connections 5, 24 and 35.

**Table 1.** Summary of the computational times for Case 2.

|  | CPU (1 Core) | CPU (6 Cores) | GPU (Tesla C2075) | GPU (Titan Black) |
| --- | --- | --- | --- | --- |
| Comp. time (h) | 30.9 | 5.43 | 0.766 | 0.264 |
| Speed-up | - | 5.7 | 40.3 | 117.1 |

The aforementioned artificial lake implies a possible flow inlet depending on the water level of the river to which it is connected. An additional injection of water discharge together with two pollutants (TSS and lead) is set in the north part of the domain (see Figure 10 (lower)). The urban area is provided with a complex drainage network with 152 ground nodes and 147 manholes, shown in Figure 10 (lower). A sewer water and TSS inflow is also set at node N8. Figure 11 shows the water discharge and concentration for both pollutants at the surface boundary inlet point (left), the stage–discharge rating curve used for the river connection (right) and the water discharge and TSS concentration at the sewer node N8 (lower). This variety of terrain types lead to the necessity of considering a complex roughness map (see Figure 12).

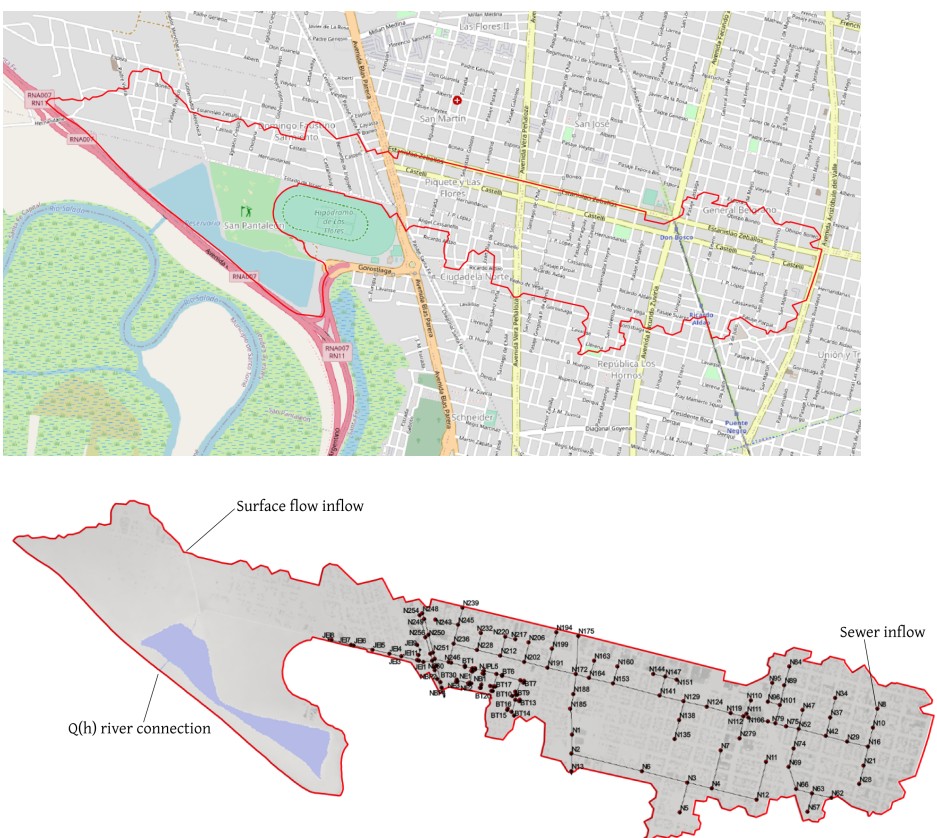

**Figure 10.** Case 3: Map of the considered area with the domain limitation (**upper**) and position of the 147 surface-sewer links (**lower**). Surface boundary conditions and sewer inflow is also depicted.

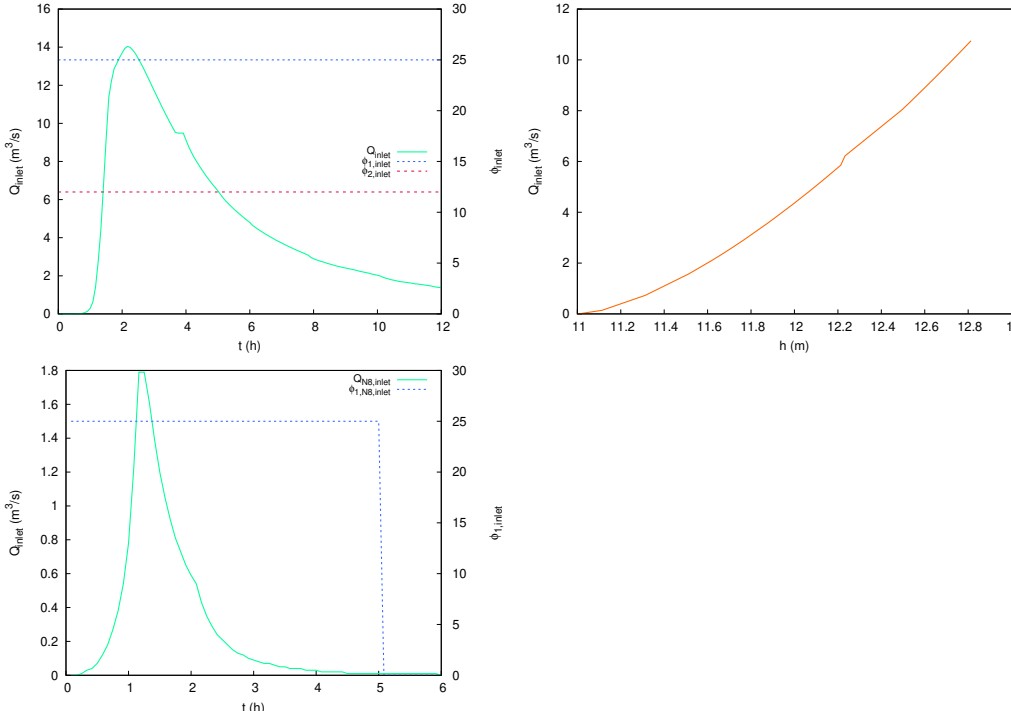

**Figure 11.** Case 3: Water discharge and concentration for both pollutants at the surface boundary inlet point (**left**). Stage discharge rating curve used for the river connection (**right**). Water discharge and TSS concentration at the sewer node N8 (**lower**).

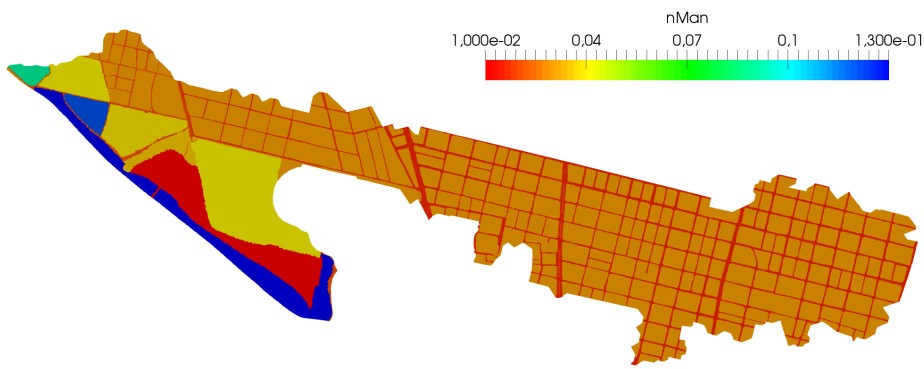

**Figure 12.** Case 3: Manning's roughness map.

Regarding the hydrological setup, an evenly distributed rainfall is considered all over the domain according to the hyetograph depicted in Figure 13. A pervious soil is considered with 11 regions with different soil characteristics which will lead to different infiltration capacities (Figure 14).

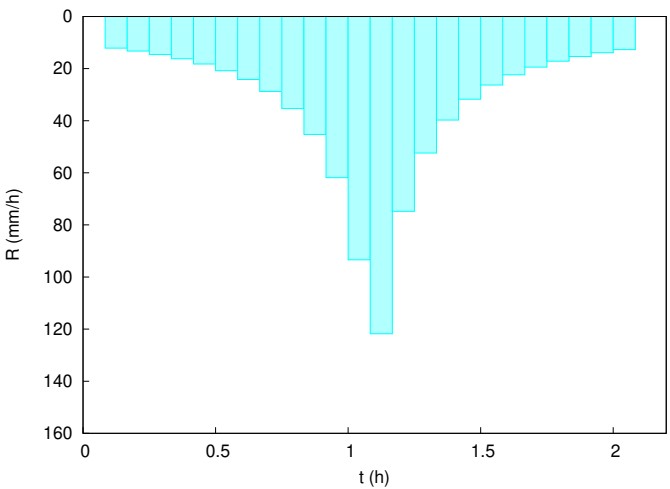

**Figure 13.** Case 3: Temporal evolution of the incoming rainfall intensity.

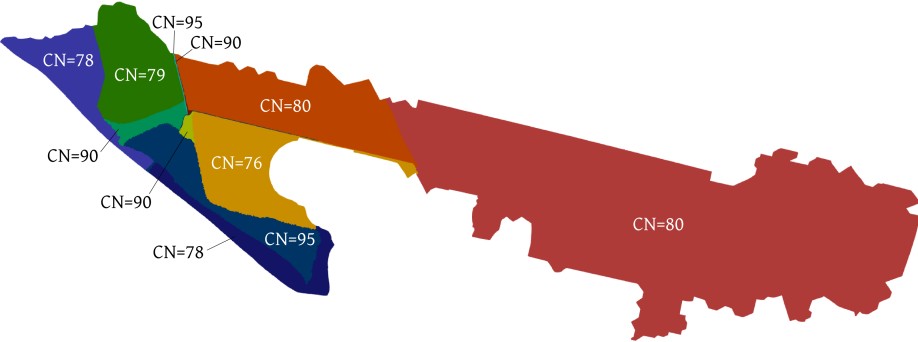

**Figure 14.** Case 3: SCS Curve Number values (CN).

The bed level $z$ of the overland domain and the initial water level for the lake and its upstream and downstream connections are shown in Figure 15. The spatial discretization is performed by means of an unstructured triangular and locally refined mesh of aproximately 600,000 elements. The mesh refinement is especially careful in the urban area, where the narrowness of the streets requires the use of small triangles in order to obtain an adequate representation (Figure 16).

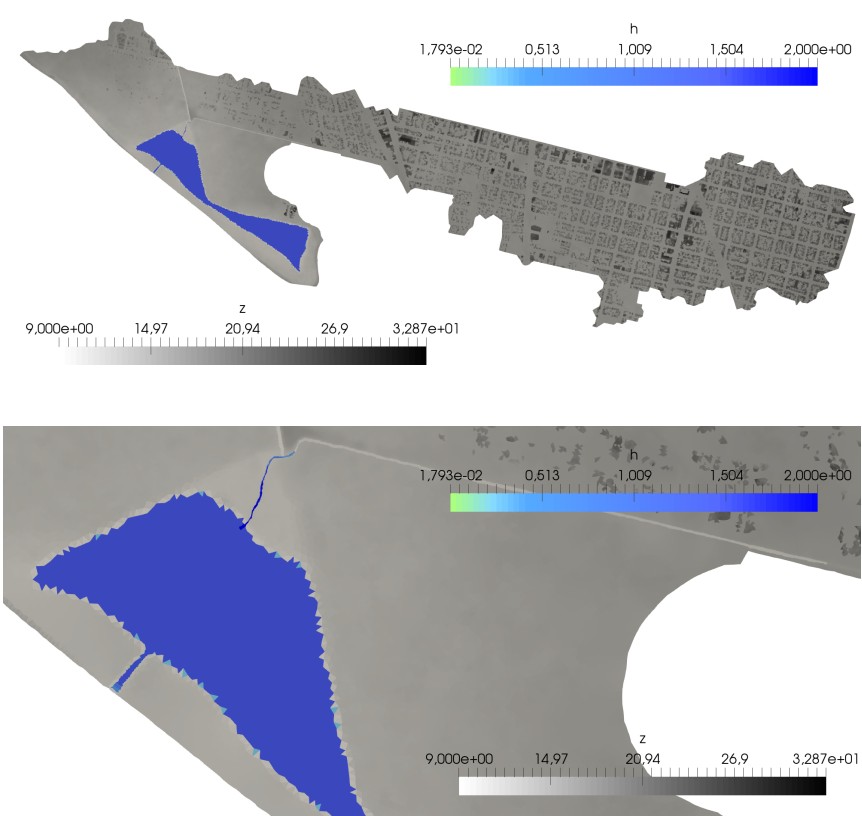

**Figure 15.** Case 3: Bed level (gray scale) and initial water depth (color scale) of the full domain (**upper**) and detail of the lake (**lower**).

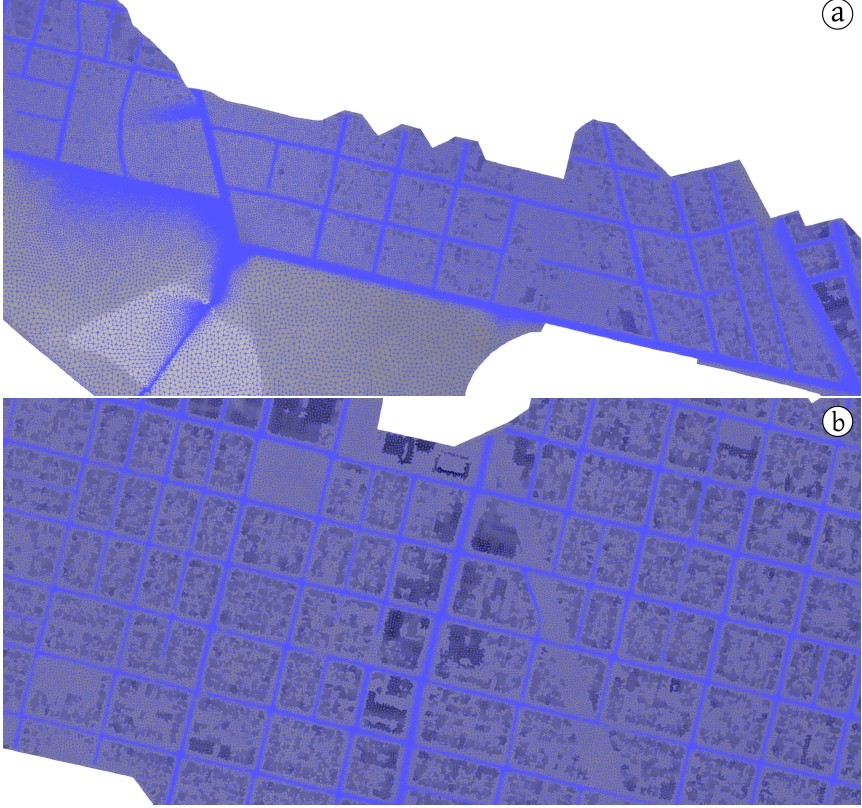

**Figure 16.** Case 3: Detail of the unstructured triangular and locally refined computational mesh (**a,b**).

Figures 17–21 show the spatial distribution of the surface flow numerical results at $t = 1.1$ h, $t = 1.5$ h, $t = 1.8$ h, $t = 5.2$ h and $t = 12$ h, respectively, for water depth $h$ and $\phi_1$ and $\phi_2$ pollutant concentrations. Figure 17 show the progression of both inlet contaminants along the northern channel and the surcharge of the sewer system causing the polluted water to reach the urban area. At $t = 1.5$ h (Figure 18) the pollution has reached the lake, spreading on it due to the water dynamics. In addition, the city supply channel is also affected. The detail shown in Figure 19 at $t = 1.8$ h reveals the drainage of the water and TSS pollutant ($\phi_1$) from the sewer to the surface in the supply channel moments before the mixing of both flows. Figure 20 shows a very advanced progression of pollutants in the lake and in the supply channel at $t = 5.2$ h, whereas Figure 21 reveals the final state of the simulation at $t = 12$ h. Several housing developments are also affected by the flood and pollutants all along the east urban area. Figure 22 shows the cumulative infiltration $F$ at the final state, defined as $F(t) = \int_0^t f\, dt$.

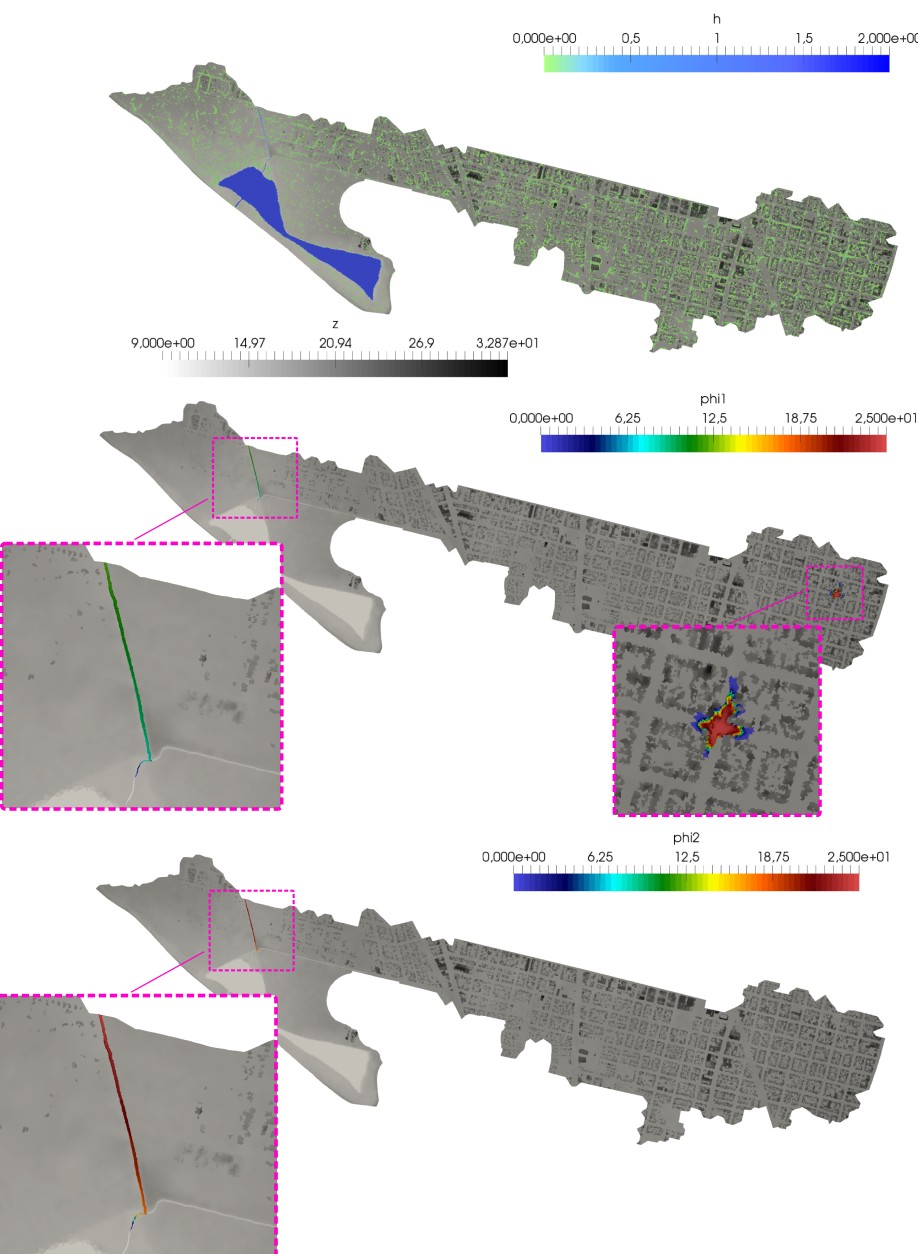

**Figure 17.** Case 3: Surface water depth $h$ (**upper**) and pollutant concentration for both solutes $\phi_1$ (**center**) and $\phi_2$ (**lower**) at $t = 1.1$ h.

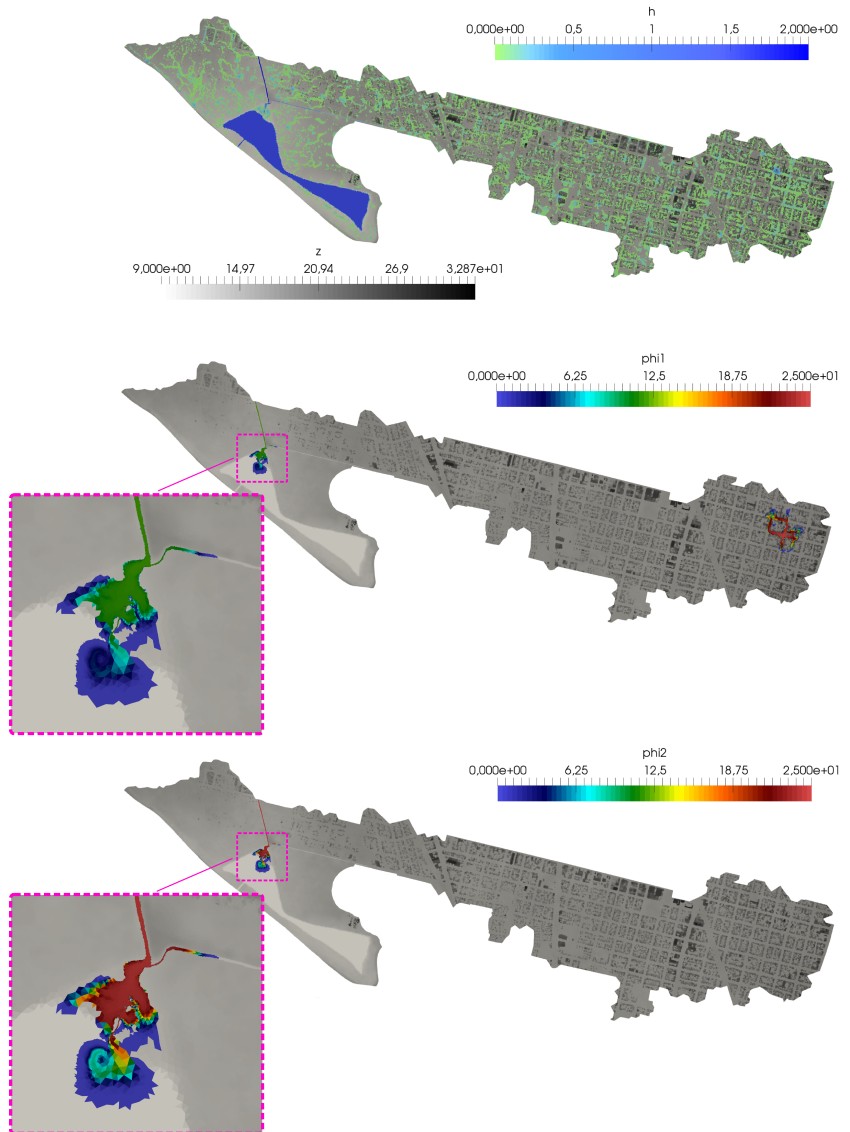

**Figure 18.** Case 3: Surface water depth $h$ (**upper**) and pollutant concentration for both solutes $\phi_1$ (**center**) and $\phi_2$ (**lower**) at $t = 1.5$ h.

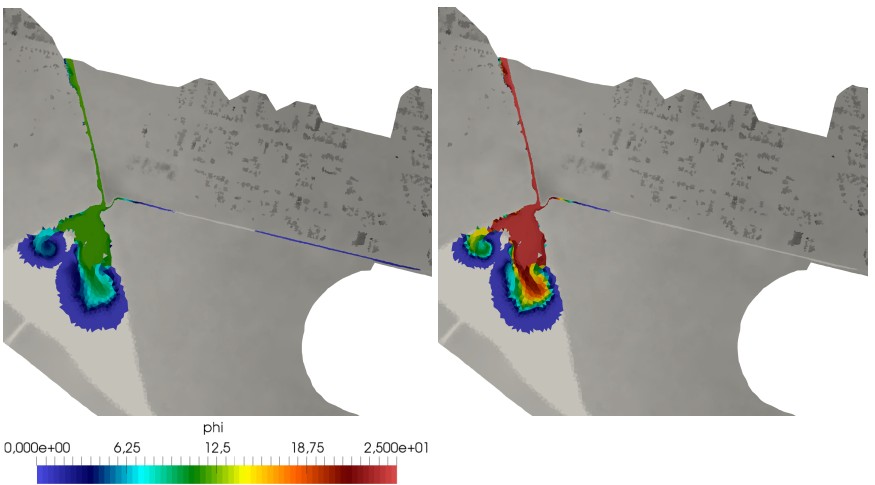

**Figure 19.** Case 3: Detail of the surface-sewer connecting channel showing the pollutant concentration for both solutes $\phi_1$ (**left**) and $\phi_2$ (**right**) at $t = 1.8$ h.

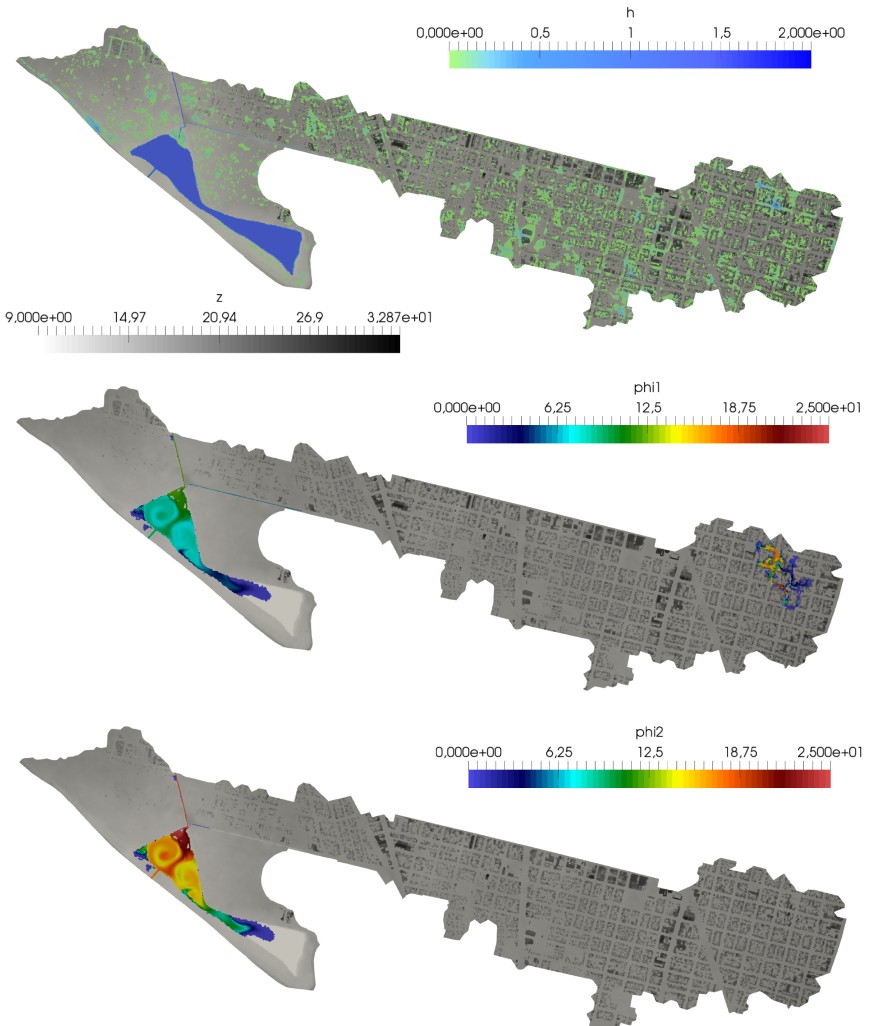

**Figure 20.** Case 3: Surface water depth $h$ (**upper**) and pollutant concentration for both solutes $\phi_1$ (**center**) and $\phi_2$ (**lower**) at $t = 5.2$ h.

As in the previous test case, a monitorization of the discharge, water depth and pollutant concentration can be carried out in every node of the sewer network. Figure 23 shows the temporal evolution of these variables at nodes N8 (upper), JEI7 (center) and N3 (lower). It is worth mentioning that the lead pollutant does not reach any point of the drainage network ($\phi_2 = 0$ for all network nodes during the full simulation). Regarding the model efficiency, the computational times and the speed-up values are shown in Table 2. Note that the magnitude of this particular case is overwhelming for a single core CPU simulation. Hence, speed-up factors are computed are the ratio between the six-core CPU time and the GPU time.

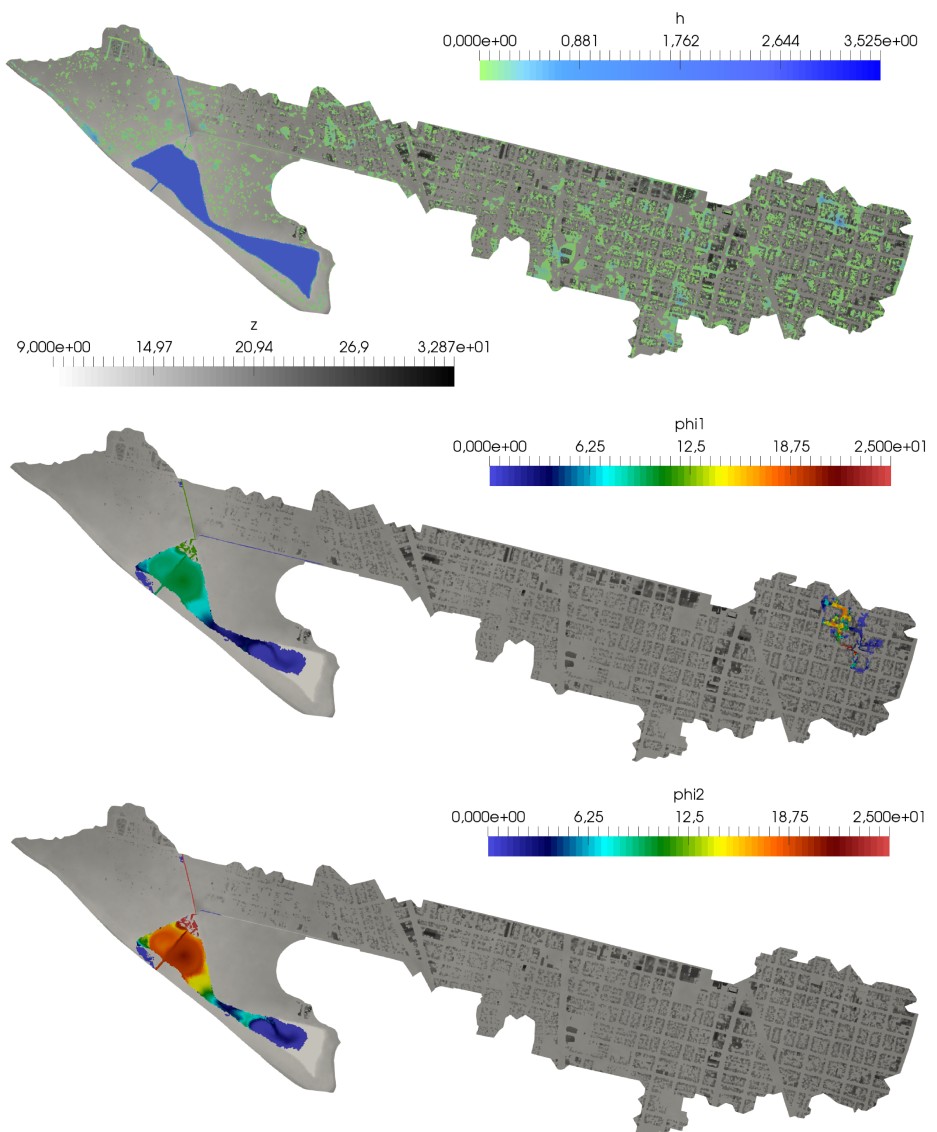

**Figure 21.** Case 3: Surface water depth $h$ (**upper**) and pollutant concentration for both solutes $\phi_1$ (**center**) and $\phi_2$ (**lower**) at $t = 12$ h.

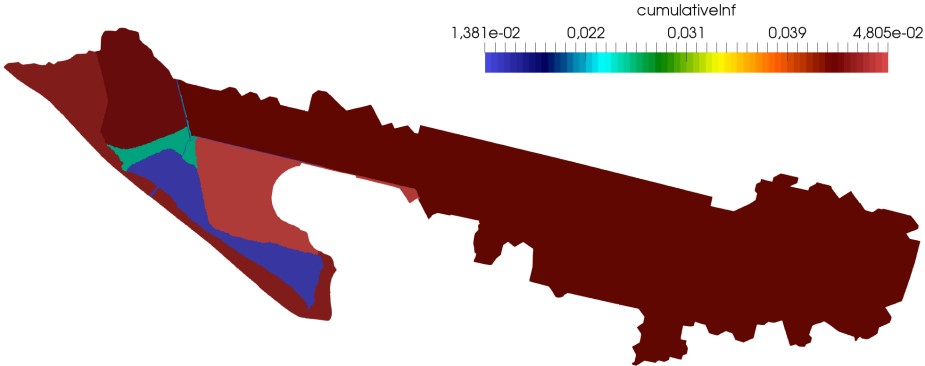

**Figure 22.** Case 3: Cumulative water infiltration $F$ at $t = 12$ h.

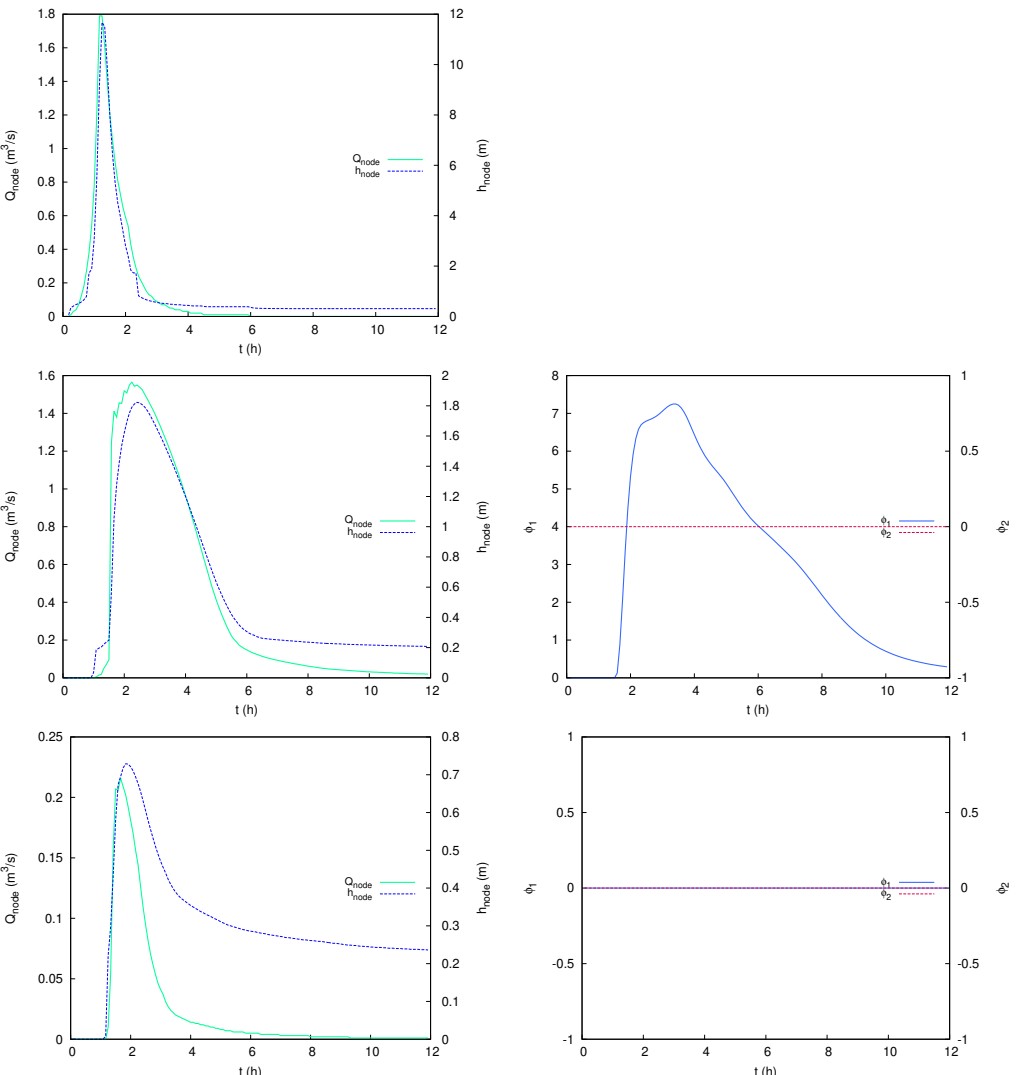

**Figure 23.** Case 3: Temporal evolution of sewer water depth and pollutant concentrations at nodes N8 (**upper**), JEI7 (**center**) and N3 (**lower**).

**Table 2.** Summary of the computational times for Case 3. In this particular case, speed-up is calculated as a factor of the 6-cores CPU time.

|  | CPU (1 Core) | CPU (6 Cores) | GPU (Tesla C2075) | GPU (GTX Titan Black) |
|---|---|---|---|---|
| Comp. time (h) | - | 135.62 | 18.05 | 5.11 |
| Speed-up | - | - | 7.5 | 26.5 |

## 4. Conclusions

In this work, the coupling of RiverFlow2D-GPU, a surface flow model based on 2D Shallow Water equations, with SWMM 5.1 software have been presented, as well as the inner computation of the bidirectional exchange flow. The passive transport of several pollutants is also considered in both models. The dynamic coupling between both domains has been carried out taking into account all the possible hydraulic scenarios. Both models are able to develop a transient water flow and pollutant transport taking into account several contaminants at the same time. Every time step, the exchange discharge and the surface and sewer water depths and pollutant concentrations are updated in order to obtain a perfect synchronization between both RiverFlow2D-GPU and SWMM flow models. Three

real-world applications at different spatial scales have been shown for validation of the model and to test the efficiency of the GPU implementation.

The first case was used to validate the flow interchange between sewer and surface by means of comparison with the numerical results provided by all the models collected by the UK Environmental Agency. The model generated numerical results consistent with those of the other models presented in [44]. Additionaly, both CPU and GPU models provide exactly the same numerical results. A very good agreement was shown as well as good GPU efficiency, despite the low number of mesh elements, which usually implies a drawback for this massive parallelization technique. The second case was used to test the ability of the model to drain the rainfall excess over a complex urban topography. In this particular case, the GPU computations reached speed-up values of 117x with respect to the single-core simulation. Finally, the third case considered a full simulation over a very large domain, taking into consideration several hydrological/hydraulic phenomena simultaneously. This setup represents a challenge for any numerical model in terms of accuracy and efficiency. The numerical results show coherence for both water flow and pollutant transport processes. It is also highly remarkable the efficiency of the GPU computation for this case, reaching speed-up values of 26.5x with respect to a high-end six-core processor. An estimated speed-up of 150x would be achieved if the simulation with a single core could be carried out.

The results and conclusions obtained in this work support the generalized paradigm shift that is taking place within the scope of hydraulic simulation. The use of GPUs against CPUs, especially in computational meshes with a large number of cells, provides significant gains in simulation efficiency. In particular, to the authors' knowledge, no previously publicated works consider either GPU computing for a fully coupled SW-based surface flow connected with a drainage model or pollutant transport calculations for this kind of model.

**Author Contributions:** J.F.-P. and P.G.-N.; methodology, J.F.-P. and P.G.-N.; software, J.F.-P.; validation, J.F.-P. and P.G.-N.; formal analysis, J.F.-P. and P.G.-N.; investigation, J.F.-P. and P.G.-N.; resources, J.F.-P. and P.G.-N.; data curation, J.F.-P. and P.G.-N.; writing—original draft preparation, J.F.-P.; writing—review and editing, J.F.-P. and P.G.-N.; visualization, J.F.-P.; supervision, P.G.-N.; project administration, P.G.-N.; funding acquisition, P.G.-N. All authors have read and agreed to the published version of the manuscript.

**Funding:** This work was partially funded by the University of Zaragoza under research project *URBAN-FLOW Desarrollo de una herramienta de simulación de flujo en zonas urbanas (VM 1/2020)*. This work has also been partially funded by Gobierno de Aragón through Fondo Social Europeo (T32-20R, Feder 2014–2020 "Construyendo Europa desde Aragón"). The authors are also thankful to Hydronia L.C.C. and to the City of Santa Fe Council for provinding the data needed to set up the third case presented in this article.

**Institutional Review Board Statement:** Not applicable.

**Informed Consent Statement:** Not applicable.

**Conflicts of Interest:** The authors declare no conflict of interest. The funders had no role in the design of the study; in the collection, analyses, or interpretation of data; in the writing of the manuscript, or in the decision to publish the results.

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
