# Peer review of "An Efficient GPU Implementation of a Coupled Overland-Sewer Hydraulic Model with Pollutant Transport"

_hydrology, doi:10.3390/hydrology8040146_

Round 1

Reviewer 1 Report

The manuscript entitled “An efficient GPU implementation of a coupled overland-sewer hydraulic model with pollutant transport” has a certain practical value. In this study, a GPU-based water quantity and quality model (RiverFlow2D-GPU) was developed and tested. However, I do not think it is suitable for publication, my reasons are as follows:

  1. The manuscript lacks a discussion or analysis of the model test results. The authors should supplement the rationality and accuracy of the simulation results. Besides, the comparison with existed or similar works should also be reflected.
  2. In the introduction (line 107-110 page 4), the authors claimed that no works consider neither GPU computing for urban hydrology/hydraulics module nor pollutant transport module. Please re-consider the current status combined with sufficient literature review and explain why so many researchers did not carry out relevant studies.
  3. In the mathematical model section, the input conditions or the basic data for the model test did not mention. Meanwhile, the framework of the coupled model and the criterion of model performance evaluation were not described.
  4. The test cases seem like a presentation of three models from the independent results of water quality, drainage flow, and GPU-based overflow. The authors did not verify the simulation results with the measured data. What is the connection among the three cases and why these cases were selected in this study?
  5. The authors need another round of very thorough proofreading. Furthermore, the citation format of the reference is not consistent with the journal requirements.

Author Response

The authors are very grateful for the thorough review and for the reviewer's comments that will undoubtedly help to improve the quality of the article.

1. The manuscript lacks a discussion or analysis of the model test results. The authors should supplement the rationality and accuracy of the simulation results. Besides, the comparison with existed or similar works should also be reflected.

ANS: To the authors knowledge, very little data are available on coupled flows between surface and sewer systems at realistic scales. For this reason, the main objective of this work is to show the relative performances of the GPU and CPU versions of the model. Therefore, the only possible comparison is the one carried out in the first test case, where the numerical results of a sewer overload are compared with those obtained by all the models compiled by the UK Environmental Agency.

2. In the introduction (line 107-110 page 4), the authors claimed that no works consider neither GPU computing for urban hydrology/hydraulics module nor pollutant transport module. Please re-consider the current status combined with sufficient literature review and explain why so many researchers did not carry out relevant studies.

ANS: As shown in Guo et al. 2021, where a comprehensive and very updated review of urban flow models is performed, there are very few coupled models that take advantage of the massive parallelization offered by a GPU. None of these GPU models are provided with a drainage network or are able to perform a water quality analysis via pollutant transport. There are many reasons that make it difficult to develop complete models in this area (computational expensiveness, requirement of good quality data input, etc.) but the authors think that the suitability (quantification of drainage floodings, simulation of urban flood dynamics with pipes, urban drainage design and evaluation, etc.) surpasses the limitations.

3. In the mathematical model section, the input conditions or the basic data for the model test did not mention. Meanwhile, the framework of the coupled model and the criterion of model performance evaluation were not described.

ANS: As stated in point 1, very little data are available on urban coupled flows between surface and sewer systems at realistic scales. Hence, the criterion of model performance evaluation is based in the efficiency of the GPU parallelization. The only possible validation is the comparison with other model results as shown in Test case 1. Regarding the pollutant transport component, no similar coupled models or field data have been found in the literature to carry out comparisons.

4. The test cases seem like a presentation of three models from the independent results of water quality, drainage flow, and GPU-based overflow. The authors did not verify the simulation results with the measured data. What is the connection among the three cases and why these cases were selected in this study?

ANS: The purpose of the three test cases is rather different. The first test performs a comparison of the hydrodynamical numerical results with the ones provided for the UK Environmental Agency which tested more than 20 hydraulic/hydrologic models in order to compare model performances. In this sense, the fact that the numerical results are contained within the envelope of the rest of the models is a good indicator that the model is working well. The second test case aims at show the GPU performance in a complex domain with a large number of computational cells. Finally, the third case, considers a realistic and very complex situation where all the hydraulic scenarios are involved in both surface and sewer network domains. As in Test 2, the huge number of computational cells makes the use of GPU paralellization mandatory.

5. The authors need another round of very thorough proofreading. Furthermore, the citation format of the reference is not consistent with the journal requirements.

ANS: The citation style has been revised and corrected according to the journal rules.

Reviewer 2 Report

This is an interesting, practical and well written paper. I enjoyed reading through the paper. Congratulations to the Authors! I do believe the paper can be published as is.

Author Response

The authors appreciate the effort made by the reviewer and the positive evaluation.

Reviewer 3 Report

  1. Line 179 ... Rubinato et al. (2017). - Publication is not describet in the literature list;
  2. Line 227 ... At t = 5h (Figure 5), ... - incorrect reference to Figure 5 instead Figure 6;
  3. Line 243 (Case 2) - too small value of the Mannings coefficient of the channels: n = 0.01.
  4. Fig. 3. - unclear description of the value on the axis z(m) in the diagram on the right.

Author Response

The authors appreciate the review anf all the comments made by teh reviewer.

1. Line 179 ... Rubinato et al. (2017). - Publication is not describet in the literature list;

ANS: The citation error has been corrected in the revised version of the manuscript.

2. Line 227 ... At t = 5h (Figure 5), ... - incorrect reference to Figure 5 instead Figure 6;

ANS: This has been corrected.

3. Line 243 (Case 2) - too small value of the Mannings coefficient of the channels: n = 0.01.

ANS: The authors agree with the reviewer. This is a small value for a typical  sewer network, but this value is not relevant for the purpose of this test case, which is to evaluate the efficiency of GPU parallelization. This has been clarified in the text.

4. Fig. 3. - unclear description of the value on the axis z(m) in the diagram on the right.

ANS: The variable is now properly describer in the figure.

Round 2

Reviewer 1 Report

The author had revised the manuscript according to the reviewer’s comments and gave a reasonable explanation one by one. I would like to suggest an acceptance of this revised manuscript for publish.